# Global Forum on Quality Assurance in CE/CPD: Assuring Quality across Boundaries

**DOI:** 10.3390/pharmacy8030114

**Published:** 2020-07-09

**Authors:** Jennifer Baumgartner, Catriona Bradley, Bronwyn Clark, Colleen Janes, Elizabeth Johnstone, Michael Rouse, Arthur Whetstone

**Affiliations:** 1Continuing Pharmacy Education (CPE) Provider Accreditation, Accreditation Council for Pharmacy Education, Chicago, IL 60603, USA; 2Irish Institute of Pharmacy, Royal College of Surgeons in Ireland, Dublin D02 FP84, Ireland; catrionabradley@rcsi.com; 3Australian Pharmacy Council, Canberra, ACT 2609, Australia; Bronwyn.Clark@pharmacycouncil.org.au; 4Canadian Council on Continuing Education in Pharmacy, St. John’s, NL A1B 1W1, Canada; Colleenj@cccep.ca (C.J.); wci@sasktel.net (A.W.); 5Professional Development, College Education and Training, Pharmaceutical Society of New Zealand, Wellington 6142, New Zealand; E.Johnstone@psnz.org.nz; 6International Services Program, Accreditation Council for Pharmacy Education, Chicago, IL 60603, USA; mrouse@acpe-accredit.org

**Keywords:** continuing education, continuing professional development, accreditation, international

## Abstract

As a result of the globalization of access and provision of continuing education and continuing professional development (CE/CPD), the national CE/CPD accreditation organizations of Australia, Canada, Ireland, New Zealand, South Africa, United Kingdom and United States formed the Global Forum on Quality Assurance of Continuing Education and Continuing Professional Development (GFQACE) to investigate and develop means of recognizing CE/CPD across boundaries. Two priorities were identified at their first meeting in 2016: (1) the development of an accreditation framework and (2) the identification of models and approaches to mutual recognition. The GFQACE approved an accreditation framework and facilitated review approach to mutual recognition in 2018 and is currently working on implementation guides. As background to the work of the GFQACE, this article provides a brief history of continuing education (CE) and continuing professional development (CPD) and discusses the value and benefits of CE/CPD to professional development of pharmacy professionals, innovation of pharmacy practice and the provision of quality patient care. Due to the essential role of CE/CPD accreditation in enabling recognition across boundaries, the nature and role of accreditation in defining, assuring and driving quality CE/CPD is described. Four conclusions regarding the broad sharing of perceptions of quality CE/CPD, the potential for expansion of the GFQACE and the benefits to pharmacy professionals, providers and pharmacy practice are discussed.

## 1. Introduction

Continuing education and continuing professional development (CE/CPD) have become increasingly important over the past five decades to pharmacy professionals as a means of maintaining their competency, enhancing their practice and providing improved patient care. In many jurisdictions CE/CPD is required by pharmacy regulatory authorities for the renewal of the license to practice. The accreditation of CE/CPD had been developed, often initiated by the pharmacy regulatory authorities, as a means of establishing quality standards for CE/CPD and ensuring the CE/CPD learning activities and programs met these standards. Currently, this accreditation has been conducted on the national and subnational levels. However, CE/CPD has increasingly become a global phenomenon with pharmacy professionals able to access CE/CPD learning activities from anywhere on the globe [1,2,3,4].

On 1 July 2016 seven national continuing pharmacy education accreditation organizations met in Split, Croatia on the day preceding the 2016 Life Long Learning in Pharmacy Conference [1]. The accreditation organizations represented Australia, Canada, Ireland, New Zealand, South Africa, United Kingdom and United States. This was the first time that these or any national continuing pharmacy education accreditation organizations met as a group, although individual organizations have shared information with each other at times over the past few decades.

At the 2016 meeting, the members shared and discussed key information about the CE/CPD accreditation system in their country, determined interest in working together to explore recognition across boundaries, identified the aim and objectives of the mutual recognition project and identified elements and steps necessary to moving forward on the consideration of recognition across boundaries. They determined that the two foundational elements needed to move forward were: (1) a commonly agreed upon set of standards or principles of quality for continuing pharmacy education accreditation and (2) models and approaches to mutual recognition of accreditation. Two work groups were established to research and prepare reports for discussion at the next meeting. Additionally, the Global Forum on Quality Assurance of Continuing Education and Continuing Professional Development (GFQACE) was established to steer and guide the work groups, share information and organize the next in-person meeting. The GFQACE would meet through videoconference between in-person meetings.

At the second in-person meeting on 6 July 2018, the two work groups presented their reports to the GFQACE [5]. The accreditation framework [6] was presented by the Accreditation Framework Work Group and the Discussion Guide: Options and Approaches to Mutual Recognition [7] was presented by the Mutual Recognition Work Group. Following extensive discussion of each report, the accreditation framework was generally accepted with the request to further clarify two of the factors (impact of CE/CPD and method of delivery) and facilitated review was accepted as the most viable option for the initial development of mutual recognition systems.

Upon achieving consensus on the requisite accreditation standards, the Accreditation Framework Work Group was tasked to develop a report for the operationalization of the accreditation framework to align with established quality indicators. The Mutual Recognition Work Group was tasked with the development of a guidance document for facilitated review. These will be presented for discussion at the next in-person meeting of the GFQACE [5].

This paper provides a background and overview of the work of the GFQACE and its achievements to date in describing an accreditation framework for CE/CPD and moving towards the mutual recognition of CE/CPD across jurisdictions. The first section provides a brief history and introduction to the two core concepts of continuing education and continuing professional development. This is followed by a discussion of the background events that led to the formation of the GFQACE. The fifth section defines accreditation and describes its role in ensuring quality CE/CPD. The sixth section provides an overview of the activities and the work of the GFQACE to date. The final two sections provide a brief discussion of issues encountered and four conclusions resulting from the GFQACE’s work.

## 2. Brief History of Continuing Education and Continuing Professional Development

The terms continuing education (CE) and continuing professional development (CPD) are conceptually distinct with, as defined in this section, continuing education referring to formal learning activities provided by a training provider and continuing professional development referring to an approach to lifelong learning that involves a broad range of learning, including informal self-directed learning and formal learning activities. However, in practice CPD is often used interchangeably with CE. As it is the formal learning activities of CPD that are typically subject to accreditation, which is the focus of this article, the authors’ use of the term CE/CPD is in order to be inclusive.

### 2.1. Continuing Education

Continuing education, referred to as further education in some jurisdictions, became very popular in the 1970s with participation tripling in the subsequent two decades [8]. The Accreditation Council for Pharmacy Education’s (ACPE) definition of continuing education reflects a common understanding of continuing education.

‘A structured educational activity designed or intended to support the continuing development of pharmacists and/or pharmacy technicians to maintain and enhance their competence. Continuing pharmacy education (CPE) should promote problem-solving and critical thinking and be applicable to the practice of pharmacy’.[9] (p.6)

From its origins in adult education dating back to the industrial revolution, there has been a strong component of vocational competency development in continuing education. In the mid-19th century government, universities and industry began to work together to educate the working class. Some of the early continuing or adult education was driven by worker movements and societies. Many universities created extension departments to extend education beyond their traditional student body. In North America, some Land Grant universities, and the equivalent in Canada, had a strong focus on agriculture training [10,11].

“Continuing education is based around the premise that students in a given area of study are looking to expand on their, already large, knowledge base. Because of it, these courses are not basic, but rather supplementary” [12]. Today, it is delivered by universities, colleges, professional organizations, private educators and a number of other educators depending on the subject. As a result, there is a wide and rich variety of courses. It is also responsive to changes in the learning needs.

Learners’ motivation to take continuing education may be voluntary and internally driven or mandatory and externally driven [13,14]. Especially since the mid-20th century, licensing and regulatory authorities for some professions like pharmacy began to require professionals to undertake continuing education as a requirement for the maintenance of their license to practice in their profession. It was viewed a means to ensure ongoing competency of the professional [12].

### 2.2. Lifelong Learning and Self-Directed Learning

Lifelong learning emerged in the 1960s and since the early 1970s has been held by major institutions as the foundational principle around which all education should be organized [15] (p. 4). By the 1990s, it was becoming a significant influence in shaping both formal and informal education and learning.

There are a number of concepts and perspectives of lifelong learning. A conventional definition of lifelong learning is:

‘… all learning activity undertaken throughout life, with the aim of improving knowledge, skills and competences within a personal, civic, social and/or employment-related perspective’[16] (p. 3)

As can be seen from this definition, lifelong learning differs from continuing education in that it has a broader scope. Further, while continuing education is often oriented toward adult education developed for the needs of schools and industries, lifelong learning focuses on the development of the potential of the whole person, recognizing each individual’s uniqueness and capacity. It involves phases of one’s life, from pre-school to post-retirement, and encompasses the whole continuum of learning from formal to non-formal and informal learning. It is viewed as a process that occurs at all times in all places.

While continuing education introduced the concept of continuing learning past formal schooling, lifelong learning introduced the concept of continuous learning driven by individual and community needs [17]. Its basic premise “is that it is not feasible to equip learners at school, college, or university with all the knowledge and skills they need to prosper throughout their lifetimes. Therefore, people will need continually to enhance their knowledge and skills, in order to address immediate problems and to participate in a process of continuous vocational and professional development. The new educational imperative is to empower people to manage their own learning in a variety of contexts throughout their lifetimes” [18] (p. 3).

A difference between continuing education and lifelong learning is that the former is usually teacher led learning while in the latter is learner driven learning. Malcolm Knowles, the father of adult learning, defined self-directed learning as follows:

‘In its broadest meaning, self-directed learning describes a process in which individuals take the initiative, with or without the help of others, in diagnosing their learning needs, formulating learning goals, identifying human and material resources for learning, choosing and implementing appropriate learning strategies, and evaluating learning outcomes’[19] (p. 2)

A self-directed learner is goal oriented, is internally motivated and more likely to undertake a learning initiative or take a course which gives them control and allows them to learn at their own pace and in their own time [19].

### 2.3. Continuing Professional Development

The concept of professional development has existed alongside continuing education since the days of Florence Nightingale. Continuing professional development developed alongside the concept of lifelong learning and incorporated many of the principles of lifelong learning [20]. Continuing professional development can been viewed as a virtuous circle of continuing competence maintenance and development.

In 2002, the Fédération Internationale Pharmaceutique (FIP) defined continuing professional development (CPD) as:

‘The responsibility of individual pharmacists for systematic maintenance, development and broadening of knowledge, skills and attitudes, to ensure continuing competence as a professional, throughout their careers’[21] (p. 2)

The FIP states that the CPD process must be “an ongoing, cyclical process of continuous quality improvement by which pharmacists seek to maintain and enhance their competence in both current duties and anticipated future service developments.” [21] (p. 2). It states that CPD must be structured and effectively managed to be effective. While there have been several variations of the CPD model since first published by FIP in 2002, all include the five essential components: self-appraisal, develop personal learning plan, undertake learning, document learning and evaluate impact on practice. The ACPE CPD model indicated that documentation needed to occur at each step of the cycle. They also added application of learning [9]. The Irish Institute of Pharmacy (IIOP) model added reflection as occurring at each step of the cycle and moving the learner to the next step [22]. These innovations in CPD are reflected in the CPD model presented below in Figure 1.

Madden and Mitchell defined CPD as meeting more than the individual professional’s learning needs. They defined CPD as being “the maintenance and enhancement of the knowledge, expertise and competence of professionals throughout their careers according to a plan formulated with regard to the needs of the professional, the employer, the professions and society” [23] (p. 134).

Continuing professional development is viewed by FIP as more than participation in continuing education, “which, on its own, does not necessarily lead to positive changes in professional practice nor does it necessarily improve healthcare outcomes” [21] (p. 2). Continuing education is one component of continuing professional development. Although the stated purpose of continuing education is to update and reinforce knowledge and skills, it is often taken simply to meet licensing requirements. It is viewed as being teacher-driven, focusing on clinical education, built primarily on education theory and using didactic learning methods, such as place-based lectures and seminars and online presentations [20].

Unlike continuing education, continuing professional development is learner-driven and can therefore be “tailored” to an individual’s learning needs. It also “builds on a broader set of theories,” [20] (p. 2) uses a “broader variety of learning methods” that include “self-directed learning and organizational and systems factors” [20] (p. 2). It focuses on clinical content, other practice-related content such as communications, human resource management and team building, pharmacy management, systems change and quality improvement. Further, “it teaches people not only how to apply solutions but also how to focus on actual performance and how to identify problems” [20] (p. 1).

## 3. Background to GFQACE

The practice of pharmacy and continuing education underwent significant change commencing in the late 1990s, gaining momentum in the early 2000s. Five trends could be viewed as the drivers that led transformation in continuing pharmacy education and the standards and practice of continuing education accreditation.

The first trend was the change in focus from continuing education to continuing professional development beginning in the late 1990s and becoming more common in the early 2000s. Continuing professional development was broader than continuing education and continuing education was viewed as one component of continuing professional development. Continuing education had been required by pharmacy regulators in many jurisdictions for maintenance of competency [24,25]. The concepts of continuing education and continuing professional development are described in the previous section [26].

The adoption of a CPD model of competence maintenance from a continuing education model by some regulatory authorities led to the introduction of concepts such as learning plans and learning portfolios as a requirement of maintenance of competence. While many jurisdictions maintained the requirement for some continuing education as a component of competence maintenance requirements, some jurisdictions dropped the requirement for continuing education and replaced it with a periodic review of learning portfolios, which were made mandatory [24]. This shift also led to the introduction of reflection and application of knowledge and skills to the standards of accreditation in some jurisdictions.

The second trend was the change in the scope of practice in pharmacy. Pharmacy practice was transformed from the dispenser of medications to a healthcare provider [27,28]. The scope of practice was expanded to include things such as prescription authority (emergency refills, extensions, refusal to administer, therapeutic substitutions, tobacco cessation, minor ailments prescribing, etc.); administration of injections and vaccines; ordering and interpreting lab tests; and conducting medication review [2].

The change of scope of pharmacy was accompanied by a change in the nature of work of pharmacy and other healthcare providers by three healthcare trends. First, there was an explosion in healthcare technologies that transformed a wide range of pharmacy and healthcare work, such as electronic records, mobile devices and wearable electronics, medical databases and big data analysis, electronic monitoring, automation, artificial intelligence, robotization and biotechnology [11]. Second, there was a change from individual practice to collaborative care and care teams involving individuals from various health professions [28,29].

The third health trend is the emergence of patient-centered care model as the centerpiece of quality healthcare. This requires listening, informing and involving the patient, and often the patient’s family, in planning and carrying out their healthcare [28,30].

The fourth trend was the transformation from the physical place-based classroom to a virtual anyplace-based classroom. This transformation was made possible by the increasing accessibility and use of digital readers and tablets and changes in Internet, social media and education technologies. The increasing speed and capacity of the Internet and mobile technologies, cloud technology, artificial intelligence, virtual reality and augmented reality are some of the main changes enabling and supporting the increase in virtual anyplace-based teaching and learning. These have been augmented by an explosion in the availability and capacity of education technologies and interactive learning management systems, authoring software, and gaming software [3,4].

The fifth trend is the internationalization of CE/CPD resulting from changes in Internet and education technologies combined with the changes in the travel industry. The significant increase in web-based learning events meant that pharmacy professionals could access CE/CPD from any organization. In addition, the increase in accessibility to conferences anywhere on the globe meant that they could also attend conferences in other jurisdictions [3,4]. In addition, the change in scope of pharmacists’ practice and increased focus on interprofessional education and practice meant that conferences initially designed and intended for other health professionals, especially physicians, were more relevant and valuable [20].

For CE/CPD accreditors, these trends led to the need to review their accreditation standards and the principles and concepts of quality that are reflected in these standards. In addition, the internationalization of CE/CPD created a challenge for the quality assurance of the CE/CPD accessed by pharmacy professionals. Accreditation organizations provide the systems and standards that the pharmacy regulatory/licensing organizations use for the assurance of quality in continuing education. To address this issue, the CE/CPD accreditation organizations of Canada, Ireland and the United States collaborated in the development of an invitational forum for national CE/CPD organizations to discuss the recognition of CE/CPD across boundaries [1].

This followed the initiative of the accreditors of continuing medical education (CME) to establish a process and guidelines to recognize the accreditation of CE/CPD from other jurisdictions. In 2002, the CME accreditation organizations of Canada and the United States established a framework for establishing substantial equivalency between continuing medical education accreditation organizations [31] (p.129). This was expanded to include ten organizations with the addition of eight European CME accreditation organizations. By 2016, there were twenty-two organizations.

## 4. Benefits of Quality CE/CPD to Patient Care and Pharmacy Professionals

Three benefits of ensuring quality CE/CPD are discussed below. Quality CE/CPD enables improved quality in patient care and improved learning, and facilitates the achievement of clinical and non-clinical performance improvement and innovation.

### 4.1. Enables Quality of Healthcare

Quality CE/CPD is viewed as important for enabling quality healthcare. First, “health professionals serve as the bridge between patients, the knowledge generated by scientific research, and the policies and practices to implement that knowledge. As the recipients of care, the public trusts health professionals to provide care that is safe, efficient, effective, timely, patient-centered, and equitable” [20] (p. 1). Evidenced-based healthcare has become a primary principle for quality patient care. This requires that healthcare providers are informed about the most current evidence. This is a major challenge in today’s world where “information has expanded at exponential rates” resulting in what is “considered to be the best knowledge one day may later be found to be inadequate” [20] (p. 1) and out of date.

Second, quality CE/CPD can assist to address a major challenge for evidenced-based healthcare. It takes 14–17 years for new evidence to be substantially implemented across the healthcare system. “Shortening this period is key to advancing the provision of evidence-based care, and will require the existence of a well-trained health professional workforce that continually updates its knowledge” [32] (p. 1). It also requires that the knowledge and skills presented and learning in CE/CPD are accurate, current and thorough. The question is: How can we ensure this?

### 4.2. Assures Improved Learning through Quality Learning Experience

Quality CE/CPD involves more than enhancing evidence-based patient care. It must also be well designed and delivered in such a manner as to promote successful learning and motivate the learner to learn. CE/CPD that is interactive, uses a variety of teaching-learning methods, involves multiple exposures, is longer and is focused on learning outcomes lead to more positive outcomes for the learner and the learner’s practice [33].

While some pharmacy professionals may engage in CE/CPD only because it is needed to satisfy licensure requirements, many are motivated by a thirst for lifelong learning to enhance their professional practice and kept abreast of new developments that help them provide optimal care. Regardless of motivation, quality CE/CPD encourages them to become self-directed learners and empowers them to take charge of their professional development and the development of their practice. While the CPD approach may require more time, effort and learning of new skills for a pharmacy professional to take responsibility for and self-manage their learning, the participation in a continuing education activity selected by using the CPD approach enhances their professional knowledge to a larger extent than choosing a continuing education activity without using a CPD approach [34].

### 4.3. Drives Performance Improvement, Innovation and Change

McMahon [35] argues that embracing CE/CPD by professionals and organizations can assist in addressing challenges in healthcare from practitioners’ well-being to better patient care and lower costs of care. It can assist in driving change and achieving goals for the professional and for their healthcare organization in both clinical areas and non-clinical areas such as quality improvement, patient safety, professionalism, team building and communication and process and system improvements.

## 5. Accreditation of CE/CPD

### 5.1. Definition and Overview of Accreditation

The standard method of assuring quality education is the accreditation of either learning activities or education providers. Accreditation in higher education in some jurisdictions dates back to the late 19th century. Accreditation of pre-service/pre-licensure pharmacy education degree programs began in early 20th century and the accreditation of continuing education programs and providers in the mid-20th century, although it has been implemented at different times in different jurisdictions.

Accreditation has been defined by the International Academy for CPD Accreditation (IACPDA) as:

‘The framework by which an educational activity is reviewed by an Accreditor or an accredited CPD provider organization to ensure the activity meets the Accreditor’s requirements and/or the process by which the Accreditor reviews and approves the organization as an accreditor’[36]

This definition notes two aspects of accreditation. First, CE/CPD accreditation may focus on either the accreditation of an individual learning activity/program or the accreditation of a CE/CPD provider organization in which the organization is authorized to accredit the learning activities or programs they deliver in accordance with the requirements established by the accrediting organization or accreditor. In some cases, the accreditor accredits another organization to accredit the learning activities of other CE/CPD provider organizations.

Second, the accreditation involves a set of requirements that the activity or program must meet in order to be accredited. These requirements may involve the content, design, development, delivery and evaluation of a learning activity, or how a CE/CPD provider does these activities, as well as the process for accreditation. They may also involve the personnel involved in the design and delivery of learning activities, the marketing of the learning activity and the capacity of the organization as well as its sustainability.

In summary, accreditation can be described as a “trust based, standards-based, evidence-based, judgment-based, peer-based process” [6] (p. 5) that results in the assurance to all stakeholders that a CE/CPD activity is a quality learning activity.

### 5.2. Why Accreditation of CE/CPD?

The drivers to establish an accreditation system by regulators and by professionals to seek accredited CE/CPD is the assurance of the accuracy of the content of a learning event, the enhancement of the effectiveness of the learning event and the minimization of commercial influence. The latter is especially important in health and pharmacy where a significant number of CE/CPD activities are sponsored by pharmaceutical and device manufacturers. Originally, the accreditation standards focused on the accuracy of the content, the nature of the organization and inputs and the delivery. Over the past couple of decades, the focus has changed from a licensing requirement to enabling optimal patient care. This change and the increasing acceptance of evidenced-based care and reflective practice has led to change in the perception of quality to require the most current, best available evidence, reflective learning and a transfer of knowledge to practice. In this section, we explore these rationales for accreditation.

#### 5.2.1. Quality Assurance—Defining, Monitoring and Driving Quality

Accreditation is one of three processes used to assure quality in higher education: quality audit, quality assessment and accreditation [37] (p. 5). Of these, accreditation is the most commonly used method.

The purpose of the accreditation standards of a national accreditation organization is to ensure the quality of CE/CPD activities. For example, the accreditation standards for the Canadian Council on Continuing Education in Pharmacy (CCCEP) state “CCCEP accreditation is designed to assure quality continuing pharmacy education learning activities and programs for all pharmacy practitioners” [38] (p. 2). The Australian Pharmacy Council (APC) states that “The accreditation of CPD activities provides an assurance to pharmacists that an activity has been reviewed for its educational quality” [39] (p. 1). The other national accreditation organizations that are members of the GFQACE have similar statements regarding their standards.

As indicated in the above statements, the standards set by the accreditation organizations frame the definition and characteristics for quality continuing pharmacy education programs. They also provide guidance to program providers and other pharmacy stakeholders. As the accreditation organization is either a pharmacy regulatory/licensing authority, which is legislated to set the standards for pharmacy education, or an organization authorized by the regulatory authority to accredit continuing education programs, the standards are the authorized criteria for quality continuing education in pharmacy.

Accreditation not only sets the standards and definition of quality and assures the quality of learning activities; it also drives quality in education and program provider organizations. The Southern Association of Colleges and Schools Commission on Colleges (SACSCOC) asserts that “accreditation requires institutional commitment to the concept of quality enhancement through continuous assessment and improvement [40]. Susan Phillips agrees that accreditation “encourages purposeful change and needed improvement” and it “guides the program or institution to continuous improvement … and addresses innovation and change” [41] (pp. 8–9).

Quality is something that is a concern to both external and internal organizational stakeholders. Learners are able to use accreditation as a guide to quality learning experiences with confidence, it assures employers of the validity of the learning activity or program, provides the profession with a concurrence on the standards of quality education for the profession, and policy makers and the public are assured that high-quality education is being provided [41] (pp. 8–9), [42]. The latter is important to the confidence of policy makers and patients in the quality of healthcare they are receiving.

#### 5.2.2. Effectiveness of CE/CPD

Not all CE/CPD is the same. The effectiveness of CPD has been challenged by a number of researchers. The CPD Standards Office in the United Kingdom, in association with the Kingston University Business School, conducted a CPD Research Project in 2010 to determine the perceived effectiveness of CPD. This project included an online survey of 1000 professionals and 40 key informant interviews of employers, academics, trainers and professional organization managers [43]. They found that the perceived quality of the CE/CPD was low and that the effectiveness of the CE/CPD was viewed as “inconsistent in its effectiveness” [43].

Manley et al. asserted that it was crucial to measure the effectiveness of CE/CPD activities; include the impact on population health; and be suitable for the teaching methodologies and learning goals. It should include the measurement of perception of satisfaction, competencies, professional performance and healthcare outcomes and provide information on whether target audience learning needs were met, stated learning objectives were achieved, participants were actively engaged, and desired behavior changes were achieved [44] (pp. 4–5).

Several researchers [44,45,46] investigated the characteristics or mechanisms of CE/CPD activities that were important to effectiveness. Some of the important mechanisms or characteristics are active, as opposed to passive, learning; a learning needs analysis; the activity reflects adult learning principles (such as autonomy, self-direction, goal orientation and practice-based learning); blended learning (mixture of online and face-to-face learning); a clear link to practice; reflective learning and grounded in research. Characteristics such as learning occurring in a collaborative environment, a supportive workplace and workplace culture are typically viewed as outside of accreditation.

One of the findings of the Standards Office study was a demand for an independent standard for determining the quality of CE/CPD activities that are relevant to “all types of CPD schemes” [43]. Filipe et al. agreed. “CPD must be amenable to external evaluation to become transparent, demonstrable, and accountable” [47] (p. 4).

#### 5.2.3. Independence and Balance of CE/CPD

Drumm, Moriarty, Rouse, Croke and Bradley argue that “In educational environments, the independence of accreditors from third-party influence is vital to ensure that the learning activities or organisations are free from commercial interests” [6]. Filipe, Silva and Stulting agree. They state that the minimization of real and potential conflicts of interest from sponsors is imperative for the validity and reliability of continuing professional development [47] (p. 5). The accreditation standards of several accreditation organizations contain provisions to restrict and minimize the introduction of bias into a learning activity by commercial interests [38,39,48].

In the 1960s in the United States, continuing education came under increasing criticism for potential conflicts of interest between pharmaceutical companies and health professionals, especially physicians. There have been several congressional investigations and hearings into this issue [20] (p. 2). This led to the regulation of continuing medical education (CME), which “began largely as a method for the American Medical Association, and eventually state medical societies, to monitor pharmaceutical influence on physician education” [20] (p. 2). This regulation of CE/CPD spread to other health professions and then to the accreditation of CE/CPD in other jurisdictions.

#### 5.2.4. Manageable Measure of Maintenance of Competence

One of the challenges for pharmacy regulatory authorities is how to ensure the competence of pharmacy professionals in an affordable and manageable way. Their mandate is to ensure that the services provided by pharmacy professionals to patients is based on the most current, best available evidence and that the care provided patients is in accordance with best practice. Public trust and confidence in the profession is dependent on the regulatory authority fulfilling this mandate [49,50].

For most regulatory authorities, it is not feasible for them to monitor every CE/CPD activity that is undertaken by licensed pharmacy professionals in their jurisdiction. The accreditation of CE/CPD provides them with a feasible and affordable means of being assured of the quality of the CE/CPD undertaken by licensees.

## 6. Global Forum on Quality Assurance in Continuing Education (GFQACE)

### 6.1. Introduction to GFQACE

The Global Forum on Quality Assurance in Continuing Education (GFQACE) was initiated at an Invitational Forum on Continuing Education Accreditation that was held on 1 July 2016 in Split, Croatia, organized to coincide with the Life Long Learning in Pharmacy Conference. Representatives from seven national accreditation organizations attended the Forum. These were the following (See Appendix A):Accreditation Council for Pharmacy Education (ACPE), United States;Australian Pharmacy Council (APC), Australia;Canadian Council on Continuing Education in Pharmacy (CCCEP), Canada;Irish Institute of Pharmacy (IIOP), Ireland;Pharmaceutical Society of New Zealand (PSNZ), New Zealand;Royal Pharmaceutical Society (RPS), United Kingdom;South African Pharmacy Council (SAPC), South Africa.

The group examined the trends and drivers impacting CE and recognition of CE across jurisdictions and identified four priority items for further discussion:Globalization and recognition of accredited continuing education across boundaries;Monitoring of quality assurance;Separation of duties between regulatory authorities and accreditation organizations;Accreditation models.

The following aim was adopted for the GFQACE: Mutual recognition of the accreditation systems based on the quality of the accreditation system.

Two priorities were identified as necessary first steps in achieving this aim and a work group was established to research each priority area and prepare a report to be tabled and discussed at the next Forum in 2018. The two priority areas were:Models and approaches of mutual recognition;Quality in continuing education.

Each work group prepared a report that was discussed at the 2018 Forum held in in Brisbane, Australia on 6 July 2018, again organized to coincide with the Life Long Learning in Pharmacy Conference. It was attended by representatives of the seven member organizations. These two reports were the initial outputs of the GFQACE. An overview of these two reports is presented in the next two sections:Discussion Guide: Models and Approaches to Mutual Recognition;Accreditation Framework.

### 6.2. Models and Approaches to Mutual Recognition

The Mutual Recognition Work Group identified three approaches by which two or more entities may recognize the standards of each other:(1)Reciprocity is the mutual exchange of rights, privileges or obligations between two or more entities.(2)Harmonization (of standards) is when the differences between the standards are minimal and minor, in technical content and application.(a)Equivalent/Identical standards are those that usually have the same title or text, and are thus identical. The presentation is word for word except for editorial changes that have no impact on meaning and application.(b)Substantial Equivalency exists when two accreditation standards and systems are deemed equivalent based on a set of agreed upon shared principles and values, which are carried through and operationalized accordingly, resulting in a significant amount of commonality and allowing for and accepting some differences in the standards and system.(3)Mutual Recognition is an inter-organizational agreement by which two or more organizations agree to recognize one another’s standards and assessments of conformity (or compliance) to those standards as being equivalent or substantially equivalent.

In this context, five models of recognition were identified and described. They are arranged in a continuum of recognition from no recognition to full recognition. Inversely related to the extent of recognition is the degree of independence of the accreditation reviews by the accreditors. As full recognition is achieved, the independence of the accreditation reviews becomes a single review.

Each of the models for recognition was described in accordance with a number of factors: recognition of system, accreditation decision making and process, impact on accrediting body and program provider, implications for stakeholders and learners, and necessary requirements or conditions to implement the option. The five models or options of recognition were (See Figure 2):Independent Accreditation. This is the current system. There is no acknowledgement that an activity or provider has been accredited by another accreditor. Applicants must go through the same application and review process as all other activities or providers.Facilitated Review. Accreditor organizations provide an expedited and simplified accreditation for activities or providers that have been accredited by another accreditor with whom they have entered into a facilitated review memorandum of understanding (MOU) and whose accreditation system meets a mutually agreed upon set of accreditation minimum standards and principles.Joint Accreditation. Accreditor organizations provide a unified application process where applicants submit one application to all parties who entered into a joint accreditation memorandum of understanding (MOU). The application may be reviewed and decided on by a joint accreditation team or each accreditor conducts their own review and decision on the accreditation of the provider or activity. Each party may review the accreditation standards of the other parties to the agreement or the parties may be required to meet a mutually agreed upon set of accreditation minimum standards and principles.Substantial Equivalency. Accreditor organizations agree on a set of principles, values and criteria and recognize the reviews of the other accreditors whose systems are deemed to meet the principles, values and criteria. An accreditor periodically submits their accreditation standards and system for review. Accreditation systems that meet the minimum requirements are seen as substantially equivalent. The CE/CPD credits earned by a pharmacy professional completing an activity that was accredited using the accreditation system of a successfully equivalent accreditor are recognized as meeting professional licensing and certification requirements of the accreditor in their jurisdiction.Harmonized Standards. Accreditor organizations agree on a set of harmonized accreditation standards and requirements and each adopts these standards and requirements. Each accreditor reviews and accredits a provider or activity using the common, harmonized standards. As activities and providers are accredited using the same standards, CE/CPD credits earned in any participating jurisdiction are fully recognized by all other jurisdictions who are part of the accreditation standards agreement. Changes to the standards require the agreement of all or a large majority of the parties to the agreement.

Following extensive discussion of the options for recognition, it was decided that the Facilitated Review was the most feasible option and best first step. It would reduce the resistance of stakeholders and provide the organizations with experience in mutual accreditation and the impact on the receiving accreditation organization. There would also be an opportunity to review and refine any special conditions required to address any issues with the delivery of the learning activities of the providers whose activities were accredited through a facilitated review. A guide to facilitated review was seen as important to enable the implementation of the facilitated review agreements between accreditor organizations.

### 6.3. Accreditation Framework

The accreditation framework was based on the work of Sarah Drumm undertaken for her thesis for the fulfillment of her Master of Science degree with the School of Postgraduate Studies, Faculty of Medicine and Health Sciences, Royal College of Surgeons in Ireland. Her research provided the evidence base for the development of the accreditation framework. Information on the framework and its development may be found in her article entitled “The development of an accreditation framework for continuing education activities for pharmacists” [6].

The framework was developed under the guidance of the Accreditation Framework Work Group of the GFQACE and her thesis supervisors Dr. C. Bradley, Prof. D. Croke, Dr. F. Moriarty and Mr. M. Rouse. The framework was intended to provide guidance to organizations and countries for the accreditation of learning activities and enable the mutual recognition of CE/CPD by two or more jurisdictions.

The accreditation framework was based on a review of the literature on “accreditation, continuing education, continuing professional development and quality” [6] (p. 4) and a survey of the GFQACE members. The quality education criteria in the accreditation framework, by which learning activities would be assessed, were significantly informed by the global framework for quality assurance in pharmacy education developed by the Fédération Internationale Pharmaceutique (FIP) as outlined in Quality Assurance for Pharmacy Education: A Global Framework [51]. The FIP Quality Framework was developed for post-secondary degree programs, and the general principles apply to CE/CPD. The aim of this framework was to “provide a structure for educational quality” [6] (p. 4) and outline the “key principles and elements that should be included in the accreditation process” [52] (p. 13).

The accreditation framework is distinct and different from the FIP Quality Framework. The FIP Quality Framework sets out the core components and requirements for quality education. The accreditation framework focuses on the accreditation system. It identifies the mechanisms and requirements of an accreditation system to ensure that the quality standards on which CE/CPD activities and providers are accredited reflect the quality education components of the FIP Quality Framework.

In developing the accreditation system elements of the accreditation framework, Drumm used the themes and patterns identified from a questionnaire of the GFQACE members and the supporting documentation they provided, along with results of the literature review and the FIP Quality Framework. Her goal was to develop a framework that reflected the accreditation processes of the organisations and was “aligned with best practice” [6] (p. 4).

Drumm used a Delphi process to determine the perspectives of the organizations in the GFQACE and develop a consensus on requisite accreditation standards. There were three rounds of the Delphi, followed by a workshop with the GFQACE members in which the results of the Delphi were presented and discussed. There was a general consensus of support for the framework. It was determined that two items needed further clarification. This was achieved by a fourth Delphi focusing on these two elements. Following a discussion and general acceptance of the framework, the accreditation framework was finalized by Drumm and the Accreditation Framework Work Group and became the foundation for the next step, an implementation and operational report, in the development of the GFQACE Accreditation Framework.

The structure of the accreditation framework consists of four stages [6] (pp. 14–19):(a)Accreditation Inputs: this relates to the inputs into the accreditation process, i.e., the structures and resources required for the development of an activity.(b)Accreditation Process: this refers to the processes involved in accreditation of an activity.(c)Accreditation Outputs: These are the outputs from the accreditation process, rather than those from the training program (these are captured in the ‘input’ stage).(d)Quality Improvement: this is carried out after the process has been completed, but feeds into the process when it is viewed as a continuum. Quality improvement (QI) is a continuous improvement process to review, critique, and implement positive change.

Each of these stages consists of two or more components, which form the main elements or standards of quality within the framework [6], as noted below.

(a)Accreditation inputs (context for activity, accreditation standards/processes, quality processes, educational content, method of delivery, assessment approach, evaluation of activity, impact of activity and reflective practice);(b)Accreditation process (application process and application review process);(c)Accreditation output (decision and appeals process);(d)Quality improvement (review of activity and evaluation by participants).

A full description of each component may be found in Appendix B.

There was consensus that the next step would be to prepare an implementation (or operational) guide which would allow an organization’s standards and guidelines to be assessed using the accreditation framework, which is to incorporate the principles outlined in the FIP Quality Framework [51] to ensure the quality of accredited CE/CPD and promote continuous quality improvement and excellence in CE/CPD accreditation.

## 7. Discussion

The GFQACE was formed to facilitate a discussion of the accreditation standards and systems by which the equivalency of the standards and systems of different organizations could be compared, thereby enabling the recognition of CE/CPD credits and courses between the accreditation agencies of the jurisdictions. The main challenge is that CE/CPD programs and the accreditation of these programs differ because of the differences in contexts and history of CE/CPD in different jurisdictions. Phillips argues that there are opportunities as well as challenges in international accreditors working together. They must keep in mind “the important but different roles of countries and governments, of different professions, of the educators, and of the accreditors” [41] (p. 13).

The organizing group of the initial meeting of the GFQACE endeavored to minimize the impact of the cultural and historical differences by selecting members who had previously exchanged information on the accreditation of CE/CPD. While this facilitated the initial discussions and increased the potential of a successful development of agreed upon principles and standards of quality for accreditation, it may have limited the application of the quality standards and its future expansion to other jurisdictions. This issue was discussed at the initial and subsequent meetings of the GFQACE. It was felt that the benefit of having a framework that identifies core quality criteria and principles, and a potential process for recognition of the accreditation system, outweighed the cost of trying to coordinate a much larger group. It is important that core criteria and principles should be viewed as fluid so that they can be revised to accommodate the different contexts and approaches of other jurisdictions.

The facilitated review option may be viewed as the least intrusive option of mutual recognition. It recognizes the accreditation system but not the accredited CE/CPD credits. The program has to be accredited by the second accreditation organization in order for the CE/CPD credits to be recognized by the pharmacy regulatory/licensing authority. While substantial equivalency or full recognition might be the ideal option because it would provide direct acceptance of the CE/CPD credits, the fact that the recognition of CE/CPD credits to meeting licensing requirements is determined by the regulatory/licensing body and not the accreditation organization was viewed as a significant barrier to these options. It was argued that starting with the facilitated review option would create an awareness and acceptance of the quality of the CE/CPD from other jurisdictions as a base for a more direct recognition option. It was viewed as a first step in the mutual recognition journey.

The accreditation framework provided a structure for looking at the whole accreditation system including the inputs to the accreditation process, the accreditation process and quality assurance. As noted above, the individual accreditation systems of the members of the GFQACE were a significant influence in determining the final elements to be included in each of the stages of the framework. This begs the question: would the framework have some different elements if the Delphi had been conducted with accreditation organizations for a range of different professions? While a review of best practices in accreditation was undertaken as part of framework development, the GFQACE may need to review this as it seeks to expand the membership to other jurisdictions.

While there was a general consensus regarding the stages and elements of the accreditation framework, there were elements for which there was a lower level of support by some members. The differences in the accreditation systems and perceptions regarding the importance of some of the elements became evident [5,6]. These differences will need to be addressed in the development of the detailed descriptions of the criteria that will be used to review and assess each element.

The benefits of CE/CPD, as identified above, include improved patient care, improvement of clinical and non-clinical performance and innovation. Manley et al. [44] stated that the measurement of the effectiveness of CE/CPD should include not just satisfaction but also competencies, professional performance and healthcare outcomes. However, the quality improvement stage of the accreditation framework requires only an evaluation by participants. It is unlikely a participant evaluation will capture all the information identified by Manley et al. While a fuller evaluation of CE/CPD may be viewed as the ideal, members have concerns about the practicality of more extensive evaluations of the outcomes of many CE/CPD activities. Many are of short duration and focus on a specific topic [5]. The framework does meet the continuous improvement aspect of quality assurance by requiring the provider to periodically review the activity.

It is unlikely that the acceptance and adoption of the accreditation framework as a measure of the quality of an accreditation system will have a major impact on the accreditation process in most member organizations. It largely reflects the current accreditation systems. However, the general agreement to align the input (i.e., program-related) elements of the accreditation framework to the structure of the FIP Quality Framework may cause some issues in the operationalization and implementation of the accreditation framework.

The FIP Quality Framework was developed to define quality for pre-licensure degree programs. It includes structural, finance and governance factors that may be less relevant to CE/CPD. Degree programs tend to be more program driven and academic oriented, and they set a baseline requirement for licensure. CE/CPD programs and activities tend to be learner driven and oriented to an individual’s practice. This has become even more of a distinguishing feature of continuing education with the incorporation of CPD characteristics such as self-assessment, self-directed learning and reflection.

It is important to ensure that the standards of quality CE/CPD evolve as healthcare, the profession, education and learning and health technologies evolve and change. One of the best practices in accreditation is continuing improvement of the accreditation system [53]. As change happens at different times in different jurisdictions, there may be no motivation or perceived need for change in some jurisdictions with a more urgent need in others. For example, the accreditation standards in most jurisdictions changed in response to the growth of “time and space boundless and web-based CPD events” [44]. (p. 4). Similarly, changes in the scope of pharmacy practice and the normalization of reflective practice led to changes in the accreditation standards in some jurisdictions [38,39,48]. How will the accreditation organizations ensure that the elements of the accreditation framework, which are the criteria by which an organization’s accreditation are assessed, remain current?

Drumm, Moriarty, Rouse, Croke and Bradley argue that the accreditation framework not only provides a measure of quality assurance for an accreditation system and a threshold for quality, it also provides accreditors with a “mechanism for improving quality” [6] (p. 1). It can be “used for developing new standards or upgrading existing standards, with continuous improvement a core feature” [6] (p. 2). Just as the framework provides the basis for recognition of CE/CPD across boundaries, it must also provide a focal point and framework for accreditors to keep track of the impact of changes in health and education practice and technologies and trends in CE/CPD across the globe and see their impact on the perspectives of quality.

Accreditation, like any process, requires time and resources. Some may question whether the extra time and resources required to accredit CE/CPD activities is worth the expense. Is there any return on this investment? It is argued in this paper that accreditation, as a measure of quality assurance, provides a return on investment to learners, employers, providers, regulators and the public. All stakeholders are concerned with providing patient care based on the most current research evidence and established best practices. Accreditation provides this assurance. In addition, all stakeholders desire activities where there is a balanced presentation of the evidence and the absence of bias.

Professionals want a learning experience that meets their learning and professional development goals, that optimizes the potential for successful learning and that gives them the biggest return for their time and effort spent in learning. Employers and funders want the learning to result in innovation and improved practice and patient care. Regulators want an affordable and manageable way of fulfilling their mandate for ensuring professional competence. Patients want confidence that the care they receive is the best care possible that will optimize their health outcomes.

Program providers often bear the expense of paying for having a CE/CPD activity accredited, in addition to the time, expertise and effort required to develop an activity that meets the accreditation standards and guidelines. However, they are able to market their learning activities as quality learning experiences that meet the quality standards of the accreditor organization, thereby assuring it is non-biased and a quality learning experience. The non-bias factor is especially important for those activities sponsored by commercial interests. Anecdotal evidence from accreditation leaders is that some professionals will only take CE/CPD activities that are accredited.

There are three limitations to this study. First, the focus of the GFQACE is on the accreditation of CE/CPD. As a result, the work of the GFQACE has related to the formalized aspects of CPD such as continuing education as opposed to the less formal as aspects of CPD such as self-directed learning and practice research undertaken by a professional. Results and conclusions may not be applicable to the less formal aspects of continuing professional development.

Second, most of the members of the GFQACE are predominantly English-speaking jurisdictions with some commonality in their approach to the accreditation of CE/CPD. The results of their work and the application of the accreditation framework may be limited to jurisdictions with a similar culture and approach to accreditation. This limitation is somewhat ameliorated in that the FIP quality education standards are incorporated into the work of the GFQACE and the FIP is an international organization with a broad membership.

Third, the authors currently work in the field of CE/CPD accreditation. The views, perspectives and conclusions regarding CE/CPD, the accreditation of CE/CPD and the importance of quality assurance may not be generalizable to other CE/CPD stakeholders. This limitation may be somewhat ameliorated by the diverse professional experience of the authors that includes pharmacy regulation, post-secondary education, academia and CE/CPD program development and delivery.

## 8. Conclusions and Next Steps

The accreditation of CE/CPD emerged in the latter half of the 20th century together with the emergence of the requirement by pharmacy regulation and licensing authorities that a pharmacist take a minimum number of CE credits as a condition for maintenance of licensure. Accreditation was adopted as a method of quality assurance of CE/CPD. Quality assurance was viewed as critical to enabling safe, quality patient care; supporting innovation and change in healthcare; assuring a quality learning experience; assuring independence and balance of content; and optimizing the effectiveness of CE/CPD.

CE/CPD accreditation and associated accreditation standards have been nationally based with no provision for the recognition of CE/CPD accredited in another jurisdiction. The aim of the GFQACE was to evaluate and support a means of recognizing accredited CE/CPD across boundaries.

### 8.1. Conclusions

Four conclusions result from this overview of the background and work of the GFQACE to develop the tools and mechanisms needed to recognize CE/CPD across jurisdictional boundaries:(1)This first substantive initiative at a global, multiple jurisdiction discussion and collaboration on accreditation and quality assurance of CE/CPD in pharmacy has demonstrated that the perception of quality standards and principles are relatively widely shared among the accreditors of the seven jurisdictions represented in the GFQACE. This shared view of quality can provide the CE/CPD accreditor, and its stakeholders, confidence that the accreditation standards in their jurisdiction are a good reflection of what constitutes quality CE/CPD.(2)An accreditation system of CE/CPD that meets the minimum quality measures of a commonly accepted accreditation framework will be required in order for accreditors to fully recognize the CE/CPD from other jurisdictions. Further, accreditors will need to be open to fully participating in a periodic peer review of their standards and guidelines against the accreditation framework and to act to ensure their standards and guidelines meet the quality measures of the accreditation framework.(3)Pharmacy stakeholders benefit from the assurance of the quality of CE/CPD and the ability of pharmacy professionals to access quality CE/CPD from a variety of jurisdictions.(a)For CE/CPD accreditors, their accreditation standards and practices benefit from sharing information from other jurisdictions regarding trends and best practices in CE/CPD and accreditation. Their stakeholders can have greater confidence in the quality of their standards thus increasing their support for the accreditor and their standards.(b)For program providers, the international recognition of CE/CPD activities and providers opens up a much larger potential market for their offerings. This improves their potential return on investment and also makes activities with specialized content more viable, especially for those located in smaller CE/PD markets.(c)For pharmacy professionals, the international recognition of CE/CPD not only expands the accessibility to a larger variety of learning activities and enriches the content of learning activities by being developed from more diverse perspectives; it can also improve their practice by expanding their professional network to pharmacy professionals with different cultures, backgrounds and experiences of their current network.(d)For pharmacy practice, the sharing of experiences with professionals from other jurisdictions can result in these professionals and their professional association advocating for a change of standards and scopes of practice, which in turn can result in easier access to improved healthcare for patients.(4)Despite the challenges of cultures, distance, time zones and work schedules, organizations can effectively and successfully collaborate to produce key tools and mechanisms that are workable in each jurisdiction and that can ultimately enhance the professional development opportunities of pharmacy professionals in all jurisdictions.

The expansion of the GFQACE membership will require careful planning and openness in order to accommodate and include jurisdictions with greater diversity in cultures, pharmacy practice, CE/CPD systems and CE/CPD accreditation systems.

### 8.2. Next Steps

Moving forward, the primary goal for the next phase of this project is to support the development of facilitated review agreements between the accreditation organizations of two or more jurisdictions. A secondary goal is the expansion of the membership of the GFQACE beyond its current seven member organizations.

The essential next steps to achieve this primary goal are (1) the operationalization and implementation of the accreditation framework and (2) the development of a facilitated review guidance document.

The Accreditation Framework Work Group is in the process of developing an operational guide which could be used to review and assess accreditation systems for meeting minimum standards. This document will outline the components required for the development of quality CE/CPD, in accordance with the FIP Quality Framework, and design the accreditation framework in a format which can be used to assess a CE/CPD accreditation system.

The Mutual Recognition Work Group is developing a guidance document, which will help implement the facilitated review option. The purpose of the guidance document is to provide a checklist and model memorandum of understanding for accreditation organizations that wish to establish a facilitated review process for accredited continuing pharmacy education activities, programs and/or providers in their jurisdictions.

The GFQACE determined that the best way of expanding its membership and to increase the awareness of the accreditation framework was to invite interested parties outside the membership to participate in a workshop at the next Lifelong Learning in Pharmacy Conference (LLLP). In addition to increasing awareness of GFQACE and its achievements to date, a workshop would allow the GFQACE to obtain feedback on the accreditation framework from a more diverse group of CE/CPD practitioners, accreditors and regulators.

The work group reports and the workshop were scheduled for July 2020 at the 2020 LLLP Conference. This conference has been rescheduled for June 2021 due to the coronavirus pandemic. Reviews and discussions are ongoing with final approval planned for the next GFQACE meeting to be held in conjunction with the 2021 LLLP Conference in Dublin.

## Figures and Tables

**Figure 1 pharmacy-08-00114-f001:**
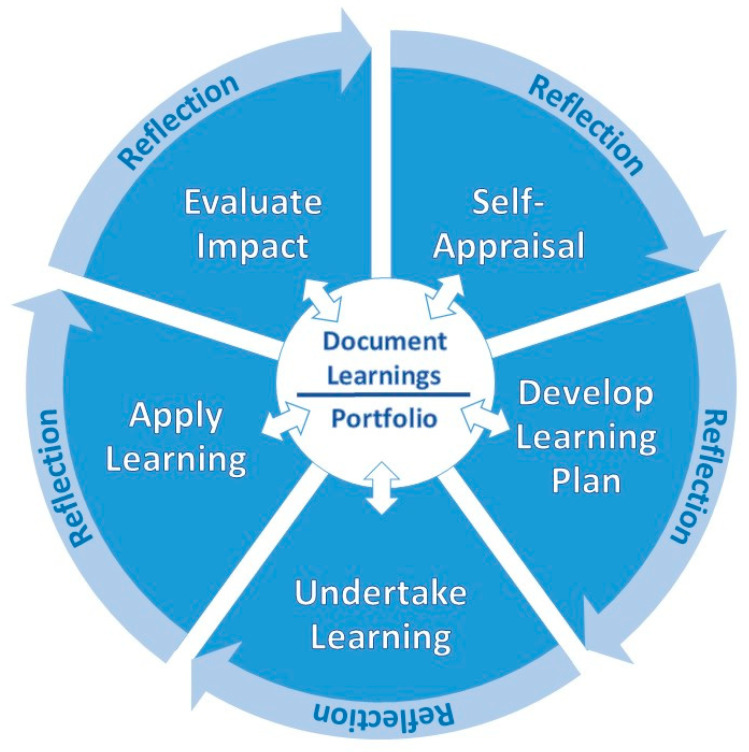
Continuing professional development process.

**Figure 2 pharmacy-08-00114-f002:**
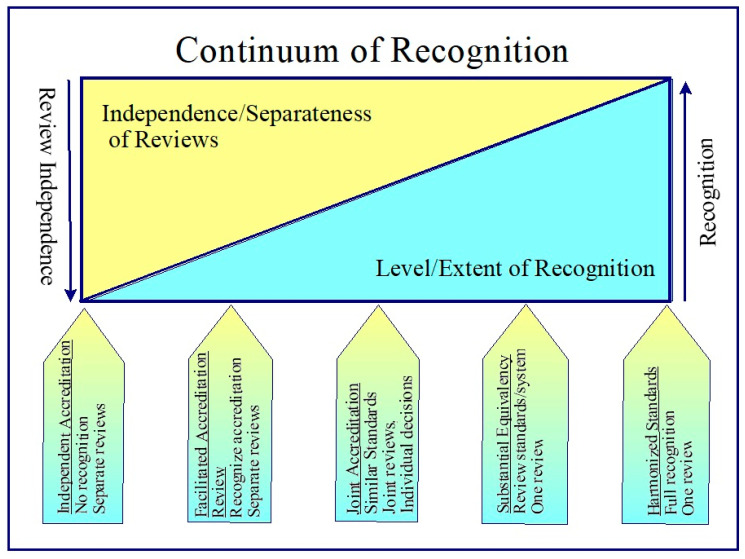
Continuum of recognition.

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
