# Peer review of "Global Forum on Quality Assurance in CE/CPD: Assuring Quality across Boundaries"

_pharmacy, 2020, doi:10.3390/pharmacy8030114_

Round 1
Reviewer 1 Report
Dear Authors,
Thanks for submitting your manuscript, I did enjoy reading it.Please find my feedback below:
- Review is organized around a coherent set of questions
- Review includes the major landmark and classic studies related to CE/CPD and accreditation processes
- Review critically evaluates the quality of the research according to
clear criteria - Review could acknowledges the authors' biases and the limitations in current accreditation processes at a higher extend
- The first paragraph in introduction needs references
- Line 56, 62 and 65: change "work groups" to "working groups"
- Paragraph starts with line 69 needs to be referenced
- Line 98: it is recommended to explain the reasons that some jurisdictions dropped the requirement for continuing education
- Paragraph starts with line 126, 134 and 137 need to be referenced
- Line 166 needs to be referenced.
- Paragraph starts with line 214 needs to be referenced
- Manageable Measure of Maintenance of Competence section needs to be referenced.
Best Wishes
Reviewer 2 Report
Thank you for the opportunity to review this manuscript. An important review and initiative for advancing accreditation of CE and CPD activities globally.
My major concerns are, acknowledging the multi-faceted aspects of accreditation of CE and CPD, focus of accreditation (eg. Individual pharmacist’s learning activities, educational offering/programmes for CE/CPD, or national legislation for practice) in the section/sentences were blurred often. The structure of the manuscript may have contributed to this issue as the definition of CE and CPD and its history were addressed in the middle of the manuscript, not at the beginning. The addressed concern also goes to a clear difference of CE and CPD, yet considered as one.
Detailed comments:
Introduction:
Line 34: A brief explanation of CE and CPD would be helpful at the beginning.
Line 55: standards or principles of quality in continuing pharmacy education – Does this mean educational programmes? Or individual practitioner’s learning activities/process? This would relate to what this group is accrediting. Could you clarify what this group is accrediting please?
Line 95: Again, what is ‘continuing education’ and ‘continuing professional development’ exactly? Some countries use one another, or sometimes confused them interchangeably. It is important to define main terms defined at the beginning.
Line 151: ‘Brief History of CE and CPD’ – It would be helpful if this section is before the background so that what CE/CPD the manuscript is talking about. I personally got confused with the perspectives of CE/CPD addressed in the manuscript, as I felt the focus of CE/CPD is more towards educational programmes and not on a self-directedness of CPD (including planned and unplanned learning/development).
Line 253: ‘Benefits of Quality CE/CPD to Patient Care and Pharmacy Professionals’ – It would be better to have this section separately as an independent section from ‘Brief History of CE and CPD.’
Line 324: The title should be corrected in a grammar.
Line 368: The section as a whole uses too many quotes from references. The majority should be paraphrased.
Line 432: Royal Pharmaceutical Society is not the accreditation organisation in the UK. Why not General Pharmaceutical Council?
Conclusion:
Line 630: A summary of review on CE/CPD, and history, and different aspects of accreditations discussed should be addressed at the beginning.
Reviewer 3 Report
This work is thorough and well-written. My main comment would be that the figures do not do a good job of adding to the article contents, and must be modified as follows:
1. The authors should redraw Figure 1 which has been adapted from IIP in a way that is original and rephrase the components of Figure 1 in their own terms.
2. Figure 2 is not coherent and does not explain how those terms are tied together. Please construct a new figure using a list/process/cycle type smartart graphic that displays those items in a more meaningful manner.
3. The discussion section is sparse, and should do a better job of tying in to the framework mentioned in the introduction, and also the future implications of the accreditation.
4. Please elaborate on future directions of the project in the conclusions section.
Round 2
Reviewer 3 Report
I appreciate the authors' efforts in revising this manuscript and can now consider it suitable for publication.